# Pigmented Paravenous Retinochoroidal Atrophy: A Case Report Supported by Multimodal Imaging Studies

**DOI:** 10.3390/medicina57121382

**Published:** 2021-12-19

**Authors:** Ilyoung Jung, Yeojin Lee, Seungbum Kang, Jaeyon Won

**Affiliations:** Department of Ophthalmology, Eunpyeoung St. Mary’s Hospital, College of Medicine, The Catholic University of Korea, Seoul 03312, Korea; iyjung.eye@gmail.com (I.J.); yeojin102@nate.com (Y.L.); john0730@hanmail.net (S.K.)

**Keywords:** pigmented paravenous retinochoroidal atrophy, bilateral, multimodal imaging

## Abstract

*Background and Objectives*: Pigmented paravenous retinochoroidal atrophy (PPRCA) is a rare disease with bilateral retinal pigment epithelium and choroidal atrophy. We present a case of PPRCA using multimodal imaging studies. *Case summary*: A 61-year-old female was referred to our department for floaters. Funduscopic examination revealed pigment clumps and grayish lesions along the retinal vein and the peripheral area, bilaterally. She did not have nyctalopia or any other visual symptoms including visual loss. She was diagnosed with pigmented paravenous retinochoroidal atrophy based on the typical findings of fundus. The findings of wide fluorescein angiography (FA), wide indocyanine green angiography (ICGA), fundus autofluorescence (FAF), spectral domain-optical coherence tomography (SD-OCT), optical coherence tomography angiography (OCTA), the visual field (VF) and an electroretinogram (ERG) could help us to confirm the diagnosis. The patient did not have any specific treatment for PPRCA in our study and there was no change in visual acuity and multimodal imaging of both eyes over one year. *Conclusions*: We report a case of PPRCA and the multimodal imaging of this patient. PPRCA is very rare disease and sometimes it is easy to get confused with other diseases such as retinitis pigmentosa and vasculitis when it comes to diagnosis. Multimodal imaging features of PPRCA will improve our understanding, diagnosis and prediction of the prognosis of this disease.

## 1. Introduction

Pigmented paravenous retinochoroidal atrophy (PPRCA) is an uncommon disease with unknown etiology characterized by retinochoroidal atrophy and pigment clumping distributed along the retinal veins, usually occurring in a bilateral and symmetric fashion [1]. It is usually asymptomatic and often diagnosed fortuitously during routine fundus examination, as the disease tends to be non-progressive or slowly progressive [2]. The cause of PPRCA remains unknown or idiopathic, but there are several hypotheses on whether this has a dysgenetic, degenerative, hereditary etiology or even an inflammatory cause [1,2].

PPRCA is rare and there are limited descriptions and diagnostic images of this disease published in the literature and case series. Recently, Lee et al. in 2021 reviewed the PPRCA cases and assessed multimodal imaging characteristics. But results for fluorescein angiography, indocyanine green angiography and optical coherence tomography angiography, a relatively novel technique, were not presented [3]. Smirnov and colleagues in 2019 reported a four-year-old boy diagnosed with PPRCA and showed clinical manifestations limited to fundus photography, fundus autofluorescence and spectral domain- optical coherence tomography of the patient [4]. In the present case, we investigated the imagery diagnostic and typical features of ultra-wide field fundus photography, fluorescein angiography (FA), indocyanine green angiography (ICGA), wide-field fundus autofluorescence, optical coherence tomography, optical coherence tomography angiography (OCTA), the visual field and electrophysiological assessment. The patient has provided informed consent for publication of the case.

## 2. Case Presentation

A 61-year old woman presented to our Department of Ophthalmology with complaints of floaters in the left eye for a week, with no complaints in the right eye. She denied remarkable systemic, ocular or familial history and was not taking any medication. She reported no symptoms of night blindness. The best corrected visual acuity was 20/20 in both eyes. Her extraocular motility and external examination were unremarkable. An anterior segment exam was normal in both eyes, without any signs of inflammatory responses of the retina and vitreous body. The intraocular pressure was 12 mm Hg in both eyes.

Ultra-wide field fundus photography revealed grayish retinochoroidal atrophic and pigment clumping lesions along the retinal vein and the fundus periphery, bilaterally. No changes were observed in both optic nerve heads and in the caliber of the retinal vessels (Figure 1A). FA showed transmitted hyperfluorescence consistent with retinal pigment epithelium (RPE) degeneration in both eyes, with more extensive areas of choriocapillaris atrophy and blocked fluorescence in the pigment accumulation areas (Figure 1B). ICGA showed a window defect with visualization of choriocapillary vessels and hypofluorescence corresponding to atrophic lesions (Figure 1C). Spectral domain optical coherence tomography (SD-OCT) of PPRCA-associated lesions documented outer retinal thinning and intraretinal hyperreflective foci with underlying shadowing corresponding to the pigment clumps. Macular SD-OCT imaging revealed no abnormal changes. The circumperipapillary retinal nerve fiber layer was preserved. Wide-field fundus autofluorescence showed increased fluorescence with a crescent-like distribution surrounding the area of RPE atrophy along the retinal vein and a decreased fluorescence pattern spreading in a fin shape into the peripheral area, bilaterally. She had normal visual evoked potentials (Figure 2A). The results of a pattern electroretinogram (ERG) were normal. A multifocal electroretinogram (mfERG) showed a markedly generalized reduced response density in the right eye and reduced amplitude paracentral areas in the left eye (Figure 2B,C). The visual field and OCTA revealed no significant abnormalities (Figure 3A,B).

The patient did not undergo any specific treatment for RPE or choriocapillaris atrophy. After a one-year follow-up, no clinical progression or changes were recorded in the fundus lesions.

## 3. Discussion

PPRCA is a rare ocular disease, typically symptomatic with peripheral retinal degeneration characterized by atrophy of the RPE, choroid and outer retinal layers. Although the diagnosis of PPRCA is based on a typical fundus appearance, the findings of FA, ICGA, fundus autofluorescence (FAF), SD-OCT, OCTA, VF and ERG can help us to confirm the diagnosis.

The wide-field fundus images showed grayish retinochoroidal atrophic and pigment clumping lesions along the retinal vein and the fundus periphery, bilaterally. This was a typical funduscopic appearance, but asymmetrical or unilateral fundus manifestations have been reported [5]. A normal optic disc and retinal vessel caliber are commonly observed in PPRCA, as observed in this case, which may help to differentiate it from retinitis pigmentosa (RP) [2]. As the patient only presented at the age of 61, there was no way of telling whether this represented progression of the retinochoroidopathy or severe disease ab initio. Nevertheless, at the one-year follow-up, no changes were recorded in the fundus lesions. Wide-field fundus images are a useful diagnostic modality and document changes that can show a variety of findings even in an early stage of PPRCA.

Lee et al. reviewed the cases diagnosed with PPRCA using multimodal imaging except for FA and ICGA. Shen et al. reported a 66-year-old female PPRCA case with FA and ICGA. They showed a window defect with visualization of medium-to-large-caliber choriocapillary vessels and hypofluorescence, respectively, corresponding to the atrophic area along the veins and the optic disc [6]. Considering PPRCA is a sporadic form of chorioretinal atrophy, we performed FA and ICGA on the patient in our case to understand more about the disease. FA of this patient showed increased transmission of fluorescence or chorioretinal atrophy, depending on the severity of the disease. No obvious fluorescein leakage was noted at any stage. Our case shows a typical FA manifestation of PPRCA in a mild form of the condition. There were multiple focal window defects with hyperfluorescence, consistent with RPE degeneration, and fluorescence blockage showed in the pigment clump lesions along the retinal vessels. ICGA showed hypofluorescence in all phases and demonstrated that hypofluorescence covered the atrophic lesions and partly extended into the areas that were hyperfluorescent with fluorescein.

Fundus autofluorescence manifestations allow spatially resolved mapping of physiological and pathological fluorophores of fundus. Hypofluorescence of FAF reveals the atrophy or disappearance of RPE because of a lipofuscin loss in RPE. And hyperfluorescence of FAF shows dysfunction of RPE that accumulates lipofuscin in RPE, meaning the possibility of RPE atrophy in the future [7]. In the case that we present, increased fluorescence with a crescent-like distribution surrounded the area of RPE atrophy along the retinal vein and a decreased fluorescence pattern spread in a fin shape into the peripheral area, bilaterally. These FAF findings could be attributed to PPRCA. Hypofluorescence documented by FAF corresponded to the areas of retinal thinning and atrophy detected by FA and OCT. SD-OCT images normally reveal thinning of retinal layers with increased backscattering and disorganization of the RPE choriocapillaris complex. On many occasions, as we saw in our case, they show severe atrophy of the choroid and RPE, with relative sparing of the inner retinal layers [8]. Hyperreflective foci with underlying shadowing corresponding to the pigment clumps were observed clinically [9]. Both the submacular choroid and central macular area maintained their integrity and regular thickness. OCTA is a novel imaging technique that resolves layer-specific microvascular details from within the retina and choroid. Shen et al. showed OCTA of their patient with PPRCA, demonstrating areas of flow void beneath the retinal pigment epithelium-Bruch membrane layer suggestive of choriocapillaris hypoperfusion that corresponded with ICGA [6]. The OCTA of the patient diagnosed with PPRCA reported by Cicinelli et al. revealed relative sparing of the retinal capillary plexuses, with diffuse defects in the choriocapillaris [10]. But OCTA research findings on PPRCA are not yet correctly defined. Our patient had normal capillary networks within the retina and choroid and we could not find explanatory signs.

The visual field showed normal-to-severe constriction—often associated with the topography of pigmentation and chorioretinal atrophy [7]. Depending on the disease status, the visual field may manifest as a ring scotoma, geographic scotoma, quadrant defect, paracentral scotomas, concentric constriction, an enlarged blind spot or scattered scotomas corresponding to the atrophic paravenous areas [1]. In our case, the visual field was normal and there was no progression in visual field loss over one year.

Electrodiagnostic data are variable and nonspecific, ranging between normal and mildly affected, to even markedly subnormal or a totally extinguished ERG [11]. This variation may signify that several conditions can present in this manner. In the case, the ERG showed an alteration in photopic and scotopic response at the periphery but with no macular involvement. A pattern electroretinogram (PERG) and the pattern visual evoked potential (PVEP) have been used in clinical studies assessing ganglion cell function in eyes with inner retina diseases [7]. Our patient had no PERG and PVEP evidence of macular dysfunction in either eye. mfERG enables topographic mapping of retinal functions within the central 40–50 degrees of the retina. It is useful for detecting local changes in retinal function that may not be detected in a full-field ERG. The mfERG showed a markedly generalized reduced response density in the right eye and reduced amplitude paracentral areas in left eye that were most severe over paracentral areas.

Based on these observations, the morphologic abnormalities in the FAF and OCT did not correlate with functional abnormalities in the mfERG. mfERG recording could be a sensitive test for detecting retinal function loss in PPRCA. The objectivity and the topographical mapping provided by mfERG enable the evaluation of localized retinal cell functional deficits in PPRCA. It is also suggested that mfERG can play an important role in early detection and monitoring of disease progression.

In the present case, the diagnosis of PPRCA was made based on a typical and characteristic fundus appearance. A multimodal approach to evaluate PPRCA using wide-field fundus photography, FA, ICGA, FAF, OCT, OCTA, VF and ERG may confirm the diagnosis and management of this disease. We recommend that ophthalmologists should take into account how a thorough examination in the diagnostic evaluation of PPRCA may improve our understanding of this extremely rare condition.

## 4. Conclusions

We have reported a case of PPRCA and the multimodal imaging of this patient. This disease is very rare worldwide and sometimes it is easy to get confused with other diseases such as retinitis pigmentosa and vasculitis when it comes to diagnosis. Multimodal imaging features of PPRCA will improve our understanding, diagnosis and prediction of the prognosis of this disease.

## Figures and Tables

**Figure 1 medicina-57-01382-f001:**
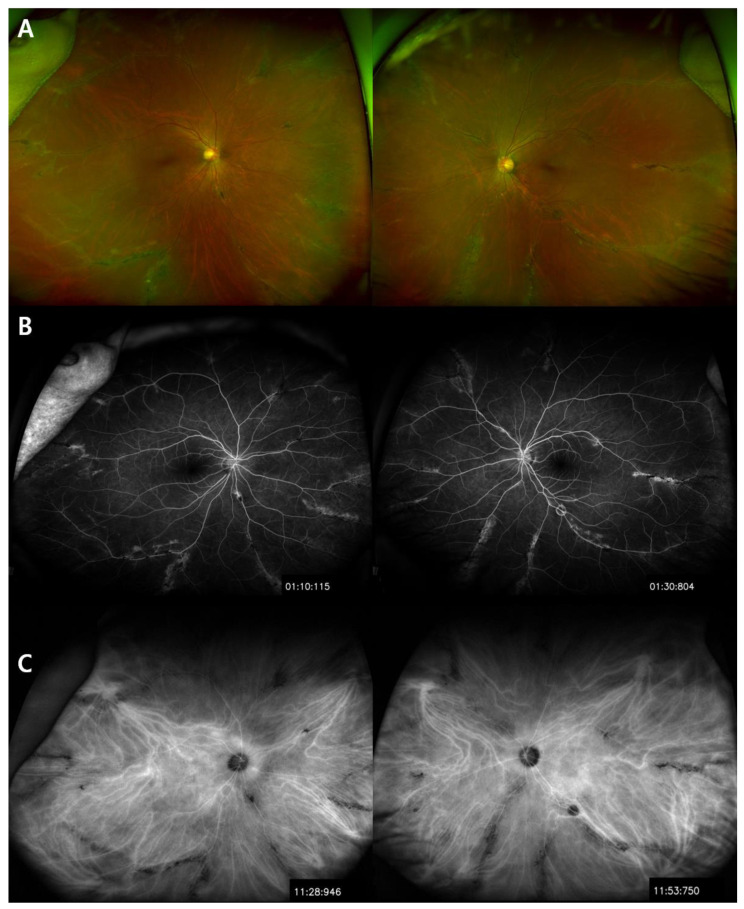
Ultra-wide field fundus photography revealed pigment clumps and grayish lesions along the retinal vein and peripheral area, bilaterally (**A**). Fluorescein angiography (FA) showed transmitted hyperfluorescence consistent with retinal pigment epithelium (RPE) degeneration in both eyes, with more extensive areas of choriocapillaris atrophy and blocked fluorescence in the pigment accumulation areas (**B**). Indocyanine green angiography (ICGA) showed a window defect with visualization of choriocapillary vessels and hypofluorescence, respectively, corresponding to the atrophic lesion (**C**).

**Figure 2 medicina-57-01382-f002:**
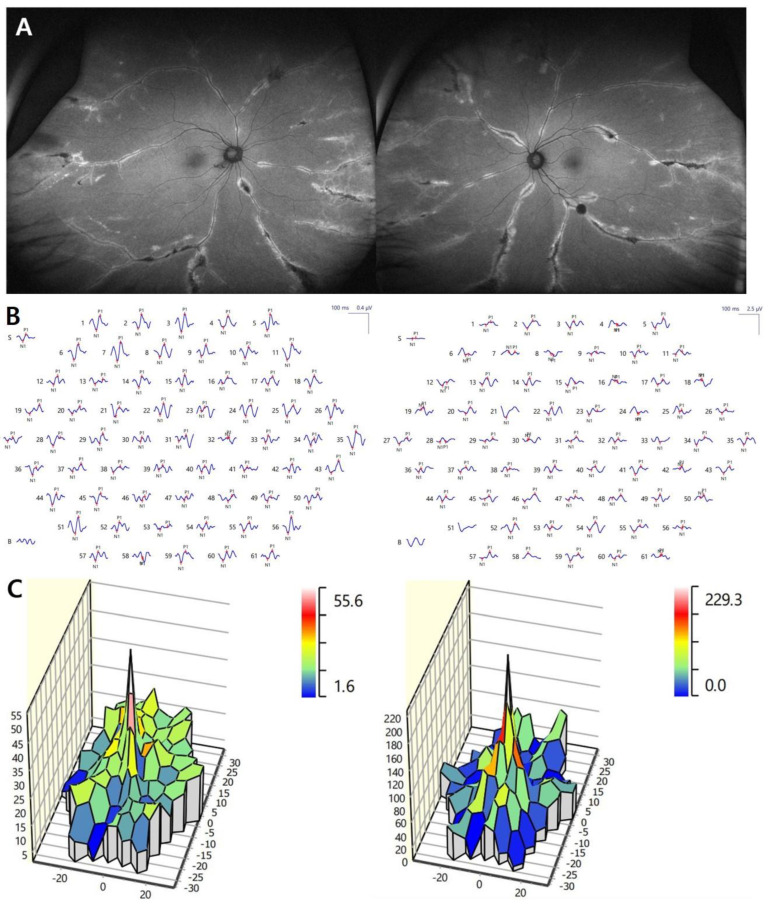
Fundus autofluorescence showed arches of increased fluorescence with a crescent-like distribution along the retinal vein and a diffuse decreased fluorescence pattern spreading in a fin-like shape into the peripheral area (**A**). There was a reduction in amplitude of all waveforms on the multifocal electroretinogram (mfERG) and markedly reduced response density in right eye (**B**). Map of P1 wave amplitudes of multifocal electroretinogram showing a reduced foveal peak in the right eye (**C**).

**Figure 3 medicina-57-01382-f003:**
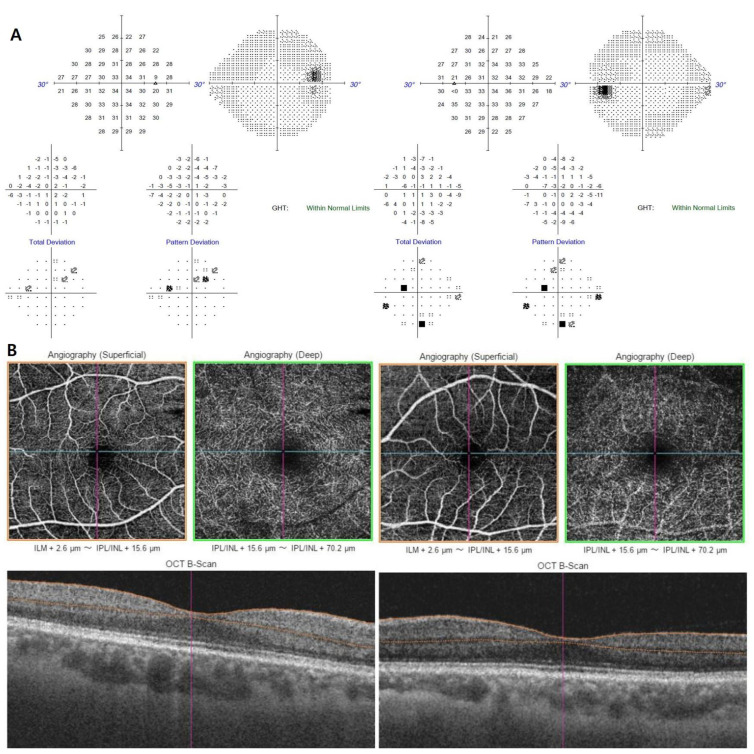
Visual field exam (SITA 24-2) showed no significant visual field defect (**A**). Optical coherence tomography angiography (OCTA) demonstrated normal capillary networks within the retina and choroid in both eyes (**B**).

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
