# Peer review of "Pigmented Paravenous Retinochoroidal Atrophy: A Case Report Supported by Multimodal Imaging Studies"

_medicina, 2021, doi:10.3390/medicina57121382_

Round 1

Reviewer 1 Report

The authors described the results of multimodal imaging for PPRCA. The findings of the imagings for PPRCA have, in fact, been reported before separately. The authors had better describe the previously reported imaging in the introduction section and compare their results with previous ones in the discussion section.

Author Response

Response to Reviewer 1 comments

Manuscript ID: medicina-1463031

Title: Pigmented Paravenous Retinochoroidal Atrophy: A Case Report Supported by Multimodal Imaging Studies

Authors: Ilyoung Jung , Yeojin Lee , Seungbum Kang , Jae Won Won *

Summary of response: We thank you for the comments and suggestions. We have carefully read your comments and found them to be extremely helpful. We believe that the manuscript has substantially improved as a result.

Point 1: The authors described the results of multimodal imaging for PPRCA. The findings of the imagings for PPRCA have, in fact, been reported before separately. The authors had better describe the previously reported imaging in the introduction section and compare their results with previous ones in the discussion section.

 Response 1: We totally agree with reviewer’s comment. We added the descriptions of two recently published reports and mentioned the differences in the introduction and the discussion section. Those two previous reports did not show the result of fluorescein angiography (FA), indocyanine green angiography (ICGA), and a relatively novel technique, optical coherence tomography angiography (OCTA). But, considering PPRCA is a sporadic form of chorioretinal atrophy, we performed FA and ICGA on the patient in our case to understand more about the disease. 

  1. Lee, E.K.; Lee, S.; Oh, B.; Yoon, C.K.; Park, U.; Yu, H.G. Pigmented paravenous chorioretinal atrophy: Clinical spectrum and multimodal imaging characteristics. Am J Ophthalmol 2021, 224, 120-132.
  2. Smirnov, V.; Ley, D.; Nelken, B.; Petit, F.; Defort-Dhellemmes, S. Pigmented paravenous chorioretinal atrophy revealing a chronic granulomatous disease. Ophalmic Genetics 2019, 40, 470-473.

Reviewer 2 Report

Dear Authors,

Manuscript ID: medicina-1463031, "Pigmented Paravenous Retinochoroidal Atrophy: A Case Report Supported by Multimodal Imaging Studies" is an atypical clinical report that clinicians and ophthalmic researchers will find intriguing. I believe that clinical photographs are a key component in making clinical reports more readable and understandable.  I'm concerned about the quality of the clinical photograph, despite the fact that the case report is well-drafted and has interesting information. Here are some of my suggestions:

  1. In the abstract, the font size for the conclusion paragraph is different. Please correct it.
  2. Please include high quality clinical photographs (for reference: 10.1016/j.ajo.2020.12.010).
  3. Some figures (Fig. 2B/C, 3A) are poorly readable, please include high-resolution images.
  4. The most recent and relevant literature is not discussed. Please address them properly in the final manuscript. Doi for relevant papers are: 10.1016/j.ajo.2020.12.010, 10.1080/13816810.2019.1681009

Best Wishes

Author Response

Response to Reviewer 2 comments

Manuscript ID: medicina-1463031

Title: Pigmented Paravenous Retinochoroidal Atrophy: A Case Report Supported by Multimodal Imaging Studies 

Authors: Ilyoung Jung , Yeojin Lee , Seungbum Kang , Jae Won Won *

Summary of response: We thank you for the comments and suggestions. We have carefully read your comments and found them to be extremely helpful. We believe that the manuscript has substantially improved as a result.

Point 1: In the abstract, the font size for the conclusion paragraph is different. Please correct it.

Response 1: Thank you for your kindness, we corrected the font size for the conclusion paragraph to match other paragraphs.

(Page 1, Line 25)

 Point 2: Please include high quality clinical photography

Response 2: Considering this manuscript is a case report, we totally agree with the reviewer’s comment. We improved the quality of the clinical photographs (Figure 1, 2, and 3).

Point 3: Some figures (Fig. 2B/C, 3A) are poorly readable, please include high-resolution images.

Response 3: Thank you for your precious comment on our manuscript. We tried our best effort to improve visibility by improving the quality of the figures.

(Fig. 2B/C, 3A)

Point 4: The most recent and relevant literature is not discussed. Please address them properly in the final manuscript.

Response 4: We totally agree with the reviewer’s comment and thank you for the priceless opinion. We added the description of the previously reported articles in the introduction and discussion section and compared them with our case report.

(Page 1, Line 42)

(Page 6, Line 115)

  1. Lee, E.K.; Lee, S.; Oh, B.; Yoon, C.K.; Park, U.; Yu, H.G. Pigmented paravenous chorioretinal atrophy: Clinical spectrum and multimodal imaging characteristics. Am J Ophthalmol 2021, 224, 120-132.
  2. Smirnov, V.; Ley, D.; Nelken, B.; Petit, F.; Defort-Dhellemmes, S. Pigmented paravenous chorioretinal atrophy revealing a chronic granulomatous disease. Ophalmic Genetics 2019, 40, 470-473.

Round 2

Reviewer 1 Report

ALL the FA, ICGA, OCTA had been simultaneously reported by Shen et al. and OCTA alone by Cicinelli et al., the authors had better mention and compare with their results instead of neglect.

  1. Shen Y, Xu X, Cao H. Pigmented paravenous retinochoroidal atrophy: a case report. BMC Ophthalmol. 2018 Jun 7;18(1):136. doi: 10.1186/s12886-018-0809-z.
  2. Cicinelli MV, Giuffrè C, Rabiolo A, Parodi MB, Bandello F. Optical Coherence Tomography Angiography of Pigmented Paravenous Retinochoroidal Atrophy. Ophthalmic Surg Lasers Imaging Retina. 2018 May 1;49(5):381-383.

Author Response

Response to Reviewer 1 comments (Round 2)

Manuscript ID: medicina-1463031

Title: Pigmented Paravenous Retinochoroidal Atrophy: A Case Report Supported by Multimodal Imaging Studies

Authors: Ilyoung Jung , Yeojin Lee , Seungbum Kang , Jae Won Won *

 Summary of response: We thank you again for the comments and suggestions. We have carefully read your comments and found them to be extremely helpful. We believe that the manuscript has substantially improved as a result.

Point 1: ALL the FA, ICGA, OCTA had been simultaneously reported by Shen et al. and OCTA alone by Cicinelli et al., the authors had better mention and compare with their results instead of neglect.

 Response 1: Thank you for your precious comment on our manuscript. We added two previously published reports' descriptions, mentioned their results, including FA, ICGA, OCTA, and compared them in the discussion section.

(Page 6, Line 114)

“Shen et al. reported a 66-year old female PPRCA case with FA and ICGA. They showed a window defect with visualization of medium-to-large-caliber choriocapillary vessels and hypofluorescence, respectively, corresponding to the atrophic area along the veins and the optic disc [6]. “

(Page 6, Line 143)

Shen et al. showed OCTA of their patient with PPRCA, it demonstrated areas of flow void beneath the retinal pigment epithelium-Bruch membrane layer suggestive of choriocapillaris hypoperfusion that corresponded with ICGA [6]. The OCTA of the patient diagnosed with PPRCA reported by Cicinelli et al. revealed relative sparing of the retinal capillary plexuses, with diffuse defects in the choriocapillaris [9].

  1. Shen Y, Xu X, Cao H. Pigmented paravenous retinochoroidal atrophy: a case report. BMC Ophthalmol. 2018 Jun 7;18(1):136. doi: 10.1186/s12886-018-0809-z.
  2. Cicinelli MV, Giuffrè C, Rabiolo A, Parodi MB, Bandello F. Optical Coherence Tomography Angiography of Pigmented Paravenous Retinochoroidal Atrophy. Ophthalmic Surg Lasers Imaging Retina. 2018 May 1;49(5):381-383.

Reviewer 2 Report

Dear Authors,

Thank you for considering my suggestions and updating with high-quality clinical photographs that look better. However, I still have reservations on Figs 2B and Fig 3A/3B (Blur and labels in the figures are not readable). Before recommending publication, I'd like to see those little revisions.

Thank You

Author Response

Response to Reviewer 2 comments (Round 2)

Manuscript ID: medicina-1463031

Title: Pigmented Paravenous Retinochoroidal Atrophy: A Case Report Supported by Multimodal Imaging Studies

Authors: Ilyoung Jung , Yeojin Lee , Seungbum Kang , Jae Won Won * 

Summary of response: We thank you for the comments and suggestions. We have carefully read your comments and found them to be extremely helpful. We believe that the manuscript has substantially improved as a result.

Point 1: I still have reservations on Figs 2B and Fig 3A/3B (Blur and labels in the figures are not readable). Before recommending publication, I'd like to see those little revisions.

Response 1: Thank you again for your thoughtful comment on our manuscript. We tried our best effort to improve visibility by improving the quality of the figures. The authors re-secured the originals of the patient's test results and retook the clinical photograph by increasing the resolution.

 (Fig. 2B, Fig. 3A/3B)

Round 3

Reviewer 1 Report

The authors have revised the manuscript appropriately.